# Optimal Solutions for Underwater Capacitive Power Transfer

**DOI:** 10.3390/s21248233

**Published:** 2021-12-09

**Authors:** Hussein Mahdi, Bjarte Hoff, Trond Østrem

**Affiliations:** Department of Electrical Engineering, UiT—The Arctic University of Norway, 8514 Narvik, Norway; bjarte.hoff@uit.no (B.H.); trond.ostrem@uit.no (T.Ø.)

**Keywords:** wireless power transfer, wireless energy transfer, capacitive coupling, network theory, underwater losses, conjugate matching, maximum efficiency, maximum power

## Abstract

Capacitive power transfer (CPT) has attracted attention for on-road electric vehicles, autonomous underwater vehicles, and electric ships charging applications. High power transfer capability and high efficiency are the main requirements of a CPT system. This paper proposes three possible solutions to achieve maximum efficiency, maximum power, or conjugate-matching. Each solution expresses the available load power and the efficiency of the CPT system as functions of capacitive coupling parameters and derives the required admittance of the load and the source. The experimental results demonstrated that the available power and the efficiency decrease by the increasing of the frequency from 300 kHz to 1 MHz and the separation distance change from 100 to 300 mm. The maximum efficiency solution gives 83% at 300 kHz and a distance of 100 mm, while the maximum power solution gives the maximum normalized power of 0.994 at the same frequency and distance. The CPT system can provide a good solution to charge electric ships and underwater vehicles over a wide separation distance and low-frequency ranges.

## 1. Introduction

Wireless power transfer (WPT) is a promising alternative to conductive (wired) power transfer for on-road electric vehicles, underwater vehicles, and ship charging applications. It has the potential to provide the convenience of automatic charging through three different modes, namely, static, quasi-dynamic, and dynamic [1]. Near-field WPT technologies are the most commonly used for high-energy charging applications. In the near-field WPT, the transmitter and the receiver sizes are much smaller, and the distance between them is much shorter than the wavelength.

The main near-field WPT approaches are inductive power transfer (IPT) and capacitive power transfer (CPT). IPT has begun to receive attention for charging electric cars, buses, and trains [2]. Many manufacturers have recently started to provide wireless charging stations with power transfer capability ranges from 3.3 up to 450 kW. However, the challenges associated with IPT are electromagnetic interference (EMI) with communication systems and human safety issues, eddy current losses, bulky size, and high cost [3,4].

CPT uses the alternating electric fields confined between the coupling plates (capacitive coupler) to transfer energy between a source and a load without physical connections. It provides a cheaper and lighter alternative to IPT with a better misalignment performance. It also addresses the IPT’s interaction problem with nearby metals and results in avoiding the eddy current losses and fire hazard potential. Consequently, it has been proposed for on-road electric vehicles [5,6,7], underwater vehicles [8,9,10,11], and ship charging applications [12,13,14].

In the electric vehicle charging applications, papers use an equivalent π-model to analyze and design air-gapped capacitive couplers [5,6,7,15]. Using this model, Orihara provides a simple formula to calculate the maximum efficiency of a CPT system using the coupling coefficient (*k*), the quality factor (*Q*), and parameters of their extended product (kQ) [16]. Dionigi et al. [17] extend Orihara’s analysis to include three possible solutions, namely, the one that maximizes the efficiency, one that maximizes the transferred power, and one that realizes power matching.

For underwater vehicles, Tamura et al. examine the efficiency of submerged CPT systems in freshwater [9,10] and seawater [11] using the same parameters that Orihara proposed. However, they do not consider the medium losses between the coupler. Moreover, they investigate the seawater at the MHz frequency range and separation distance between the couplers up to 180 mm. In contrast, Mahdi et al. study the CPT system’s maximum available efficiency and power using a conjugate matching approach and consider the dissipative losses of the seawater [14]. The main contributions of this paper are:1.It extends the analysis proposed in [14] to include three possible solutions, namely, the one that maximizes the efficiency, the one that maximizes the transferred power, and the one that realizes power matching.2.It considers the dielectric losses of seawater, unlike the previous analysis for lossless medium in [17].3.It investigates the under seawater CPT system behavior at the 0.3 to 1 MHz frequency range and separation distance up to 300 mm.

The analysis shows under which conditions each solution is achieved and demonstrates the reciprocal relationship between power and efficiency.

## 2. System Analysis

As proposed in [14], the reciprocal two-port network representation provides a general formulation for the CPT scheme. In this representation, the capacitive couplers are a black box from which only the voltages ($u_{1},u_{2}$) and the currents ($i_{1},i_{2}$) can be measured. The two-port model presents the admittance representations of the π-model in a matrix form, as illustrated in Figure 1. The system is assumed to be passive, linear, and reciprocal.

### 2.1. Passive Linear Reciprocal System

The black box model of the system contains passive components, which means there are no energy sources. Thus, the power on the receiver is less or equal to the power on the transmitter side. In addition, the system is linear as the admittance is assumed to be independent of the voltages and currents. Moreover, it is reciprocal, meaning that the transmission of the signals at any side does not depend on the direction of the propagation. The reciprocal characteristics are a result of the fundamental symmetries of Maxwell’s equations. Based on these assumptions, the admittance matrix expresses the voltage-to-current relationship as:(1)$$\left[\begin{array}{c} i_{1}\\ i_{2} \end{array}\right]=\left[\begin{array}{cc} Y_{1} & Y_{12}\\ Y_{12} & Y_{2} \end{array}\right]\left[\begin{array}{c} u_{1}\\ u_{2} \end{array}\right],$$
where $Y_{i}=g_{i}+jb_{i},\;i=1,\;2,$ or 12 is the admittance. The direction of $u_{2}$ has been reversed to achieve a positive definite matrix as noticed in Equation (Equation 1).

Figure 2a illustrates when the two-port network is connected to a source and a load. The current flows through the load is ($i_{2}=-Y_{L}u_{2}$); where the negative sign is attributed to the reversed direction of the receiver voltage ($u_{2}$). By substituting this current in the second row of the matrix, the input admittance ($Y_{\mathrm{in}}$) as it is seen by the transmitter side, can be expressed as:(2)$$Y_{\mathrm{in}}=Y_{1}-\frac{Y_{12}^{2}}{Y_{2}+Y_{L}},$$
where $Y_{L}$ is the load admittance at the receiver side. The input admittance replaces the two-port network and the load. The voltage gain across the coupler can be expressed:(3)$$G_{u}=-\frac{u_{2}}{u_{1}}=-\frac{Y_{12}}{Y_{2}+Y_{L}}$$

By substituting the source current ($i_{s}=Y_{s}u_{1}+i_{1}$) in the matrix the two port model, Equation (Equation 1), can be replaced by:(4)$$\left[\begin{array}{c} i_{s}\\ i_{2} \end{array}\right]=\left[\begin{array}{cc} Y_{1}+Y_{s} & Y_{12}\\ Y_{12} & Y_{2}+Y_{L} \end{array}\right]\left[\begin{array}{c} u_{1}\\ u_{2} \end{array}\right]$$

This matrix is an extended form of the matrix in the Equation (Equation 1) when the two-port network is connected to the source and the load. By applying an open circuit condition ($i_{2}=0$) in the second row and substituting the results in the first row, then the transmitter voltage can be expressed as a function of the source current:(5)$$u_{1}=\frac{Y_{2}+Y_{L}}{\Delta }i_{s},$$
where $\Delta =\left(\;Y_{1}+Y_{s}\;\right)\left(Y_{2}+Y_{L}\;\right)-Y_{12}^{2}$ is the determinant of the matrix in Equation (Equation 4). While Equation (Equation 3) shows that the voltage gain depends on the load, Equation (Equation 5) indicates that the current gain is independent of the load.

Similarly, the source and the two-port network can be replaced using the Norton equivalent circuit connected to the receiver side. The corresponding Norton equivalent at the receiver side, as illustrated in Figure 2b, can be expressed as:(6)$$Y_{N}=Y_{out}=Y_{2}-\frac{Y_{12}^{2}}{Y_{1}+Y_{s}}$$
(7)$$i_{N}=\frac{Y_{12}}{Y_{1}+Y_{s}}i_{s}$$

From Equations (Equation 3) to (Equation 7), the source power is expressed as: (8)Ps=12Re(Ys)u12=12gsY2+YLΔ2is2

The input power can also be expressed as: (9)Pin=12Re(Yin)u12=12ginY2+YLΔ2is2

Likewise, the power transfer to the load can be defined as:(10)PL=12Re(YL)u22=12gLY12Δ2is2

Then, the efficiency of the system can be expressed as:(11)η=PLPs+Pin=Re(YL)Re(Ys+Yin)Y12Y2+YL2

Based on the previous analysis, there are three possible solutions to design the system:(a)Maximum efficiency: determine the value of the source and the load admittance that achieve maximum efficiency.(b)Maximum power: determine the value of the source and the load admittance that achieve maximum power transfer to the load.(c)Conjugate-image: determine the values of the admittance that realize the principle of power matching.

### 2.2. Maximum Efficiency Solution

For mathematical convenience, the source is considered lossless ($g_{2}=0$), and the following expressions are defined:(12)$$\left\{\begin{array}{c} \alpha =g_{1}\\ \beta =2\;g_{1}\;g_{2}-g_{12}^{2}+b_{12}^{2}\\ \gamma =2\;g_{1}\;b_{2}-2\;g_{12}\;b_{12}\\ \lambda =g_{1}\left(\;g_{2}^{2}+b_{2}^{2}\;\right)-g_{2}\left(\;g_{12}^{2}+b_{12}^{2}\;\right)-2\;b_{2}\left(\;g_{12}\;b_{12}\;\right)\\ \zeta =g_{12}^{2}+b_{12}^{2} \end{array}\right.$$

The Equation (Equation 11) can be rewritten as:(13)$$\eta =\frac{\zeta \;g_{L}}{\alpha \left(\;g_{L}^{2}+b_{L}^{2}\;\right)+\beta \;g_{L}+\gamma \;b_{L}+\lambda }$$

In addition, defining the following two parameters for the convenience of mathematical symbols in [14]:
(14a)$$\begin{array}{cc} \psi^{2} & =\frac{g_{12}^{2}}{g_{1}\;g_{2}} \end{array}$$
(14b)$$\begin{array}{cc} \chi^{2} & =\frac{b_{12}^{2}}{g_{1}\;g_{2}}, \end{array}$$
where ψ represents the ratio between the coupling conductance and the self-conductance and χ is an equivalent to the quality factor. Another two expressions $\theta_{G}$ and $\theta_{B}$ are used, which were defined in [18] as intermediate variables, but their physical meanings are not given. The source and load admittance can be rewritten as:
(15a)$$\begin{array}{cc} \theta_{G} & =\sqrt{\left(\;1-\psi^{2}\;\;\right)\;\left(\;1+\chi^{2}\;\;\right)} \end{array}$$
(15b)$$\begin{array}{cc} \theta_{B} & =\psi \;\chi \end{array}$$

The maximum efficiency is achieved when $\partial \eta /\partial g_{L}=0$ and $\partial \eta /\partial b_{L}=0$. By solving $\partial \eta /\partial b_{L}=0$, the $b_{L}^{\left(a\right)}$ can be calculated. This paper denotes the following: superscript (a) the parameters of the maximum efficiency solution, superscript (b) those relative to the maximum power solution, and superscript (c) those related to the conjugate-image solution. Substituting the calculated value $b_{L}^{\left(a\right)}$ in Equation (Equation 13), the $g_{L}^{\left(a\right)}$ can be calculated:
(16a)$$\begin{array}{cc} g_{L}^{\left(a\right)} & =g_{2}\;\theta_{G} \end{array}$$
(16b)$$\begin{array}{cc} b_{L}^{\left(a\right)} & =g_{2}\;\theta_{B}-b_{2} \end{array}$$

The source admittance is expressed as:(17)$$Y_{s}^{\left(a\right)}=jb_{s}^{\left(a\right)}=-jb_{1}$$

The expression for the maximum efficiency is obtained by substituting Equation (16) into (Equation 13), giving:(18)$$\eta^{\left(a\right)}=\frac{\psi^{2}+\chi^{2}}{{\left(\;1+\theta_{G}\;\right)}^{2}+\theta_{B}^{2}}$$

Evaluation of ${\left|\;\Delta \;\right|}^{2}$ for a given values in Equation (16) gives:(19)$${\left|\;\Delta \;\right|}^{2}={\left(\;g_{1}\;g_{2}\;\right)}^{2}\;\left\{{\left[\;\theta_{G}\left(\;1+\theta_{G}\;\right)+\theta_{B}^{2}\;\right]}^{2}-\theta_{B}^{2}\;\right\}$$

The maximum power available from the source can be calculated from Equation (Equation 9) as:(20)$$P_{s},\max=\frac{1}{8}\frac{{\left|\;i_{s}\;\right|}^{2}}{g_{\mathrm{in}}}$$

The available power transfer to the load at the maximum efficiency can be calculated from Equations (16) and (Equation 19) into (Equation 10) as:(21)$$P_{L}^{\left(a\right)}=4P_{s},\max\frac{\left(\;\psi^{2}+\chi^{2}\;\right)\;\theta_{G}}{{\left[\;\theta_{G}\left(\;1+\theta_{G}\;\right)+\theta_{B}^{2}\;\right]}^{2}+\theta_{B}^{2}}$$

If the loses of the seawater is neglected, then the conductance, the ψ parameter, the $\theta_{G}$ parameter are zeros. Then Equations (Equation 18) and (Equation 21) can be rewritten as:
(22a)$$\begin{array}{cc} \eta^{\left(a\right)} & =\frac{\chi^{2}}{\;{\left(\;1+\sqrt{1+\chi }\;\right)}^{2}} \end{array}$$
(22b)$$\begin{array}{cc} P_{L}^{\left(a\right)} & =4P_{s},\max\frac{\chi^{2}}{\sqrt{1+\chi^{2}}{\left(\;1+\sqrt{1+\chi^{2}}\;\right)}^{2}} \end{array}$$

Equations (22) are the same for the lossless medium analysis reported in [17].

### 2.3. Maximum Power Solution

The load admittance ($Y_{L}$) that realizes maximum power transfer can be achieved in a simpler way than the procedures in Section 2.2. By using the equivalent circuit at the receiver side, which is illustrated in Figure 2b, the load admittance can be calculated using the maximum power transfer theorem. This theorem states that maximum power transfer is achieved when the $Y_{L}$ is the conjugate of the output admittance expressed in Equation (Equation 6) (i.e., when the following condition is achieved $g_{L}=g_{N}$ and $b_{L}=-b_{N}$). Thus, the value $b_{L}^{\left(b\right)}$ and the $g_{L}^{\left(b\right)}$ can be calculated:
(23a)$$\begin{array}{cc} g_{L}^{\left(b\right)} & =g_{2}\;\left(\;\theta_{G}^{2}+\theta_{B}^{2}\;\right) \end{array}$$
(23b)$$\begin{array}{cc} b_{L}^{\left(b\right)} & =2\;g_{2}\;\theta_{B}-b_{2} \end{array}$$

The source admittance can be expressed as:(24)$$Y_{s}^{\left(b\right)}=jb_{s}^{\left(b\right)}=-jb_{1}$$

The expression for the maximum efficiency is obtained by substituting Equation (23) into (Equation 13), giving:(25)$$\eta^{\left(b\right)}=\frac{\psi^{2}+\chi^{2}}{2\left(\;1+\theta_{G}^{2}+\theta_{B}^{2}\;\right)}$$

Evaluation of ${\left|\;\Delta \;\right|}^{2}$ for a given values in Equation (23) gives:(26)$${\left|\;\Delta \;\right|}^{2}={\left(\;2\;g_{1}\;g_{2}\;\right)}^{2}\;{\left[\;\theta_{G}^{2}+\theta_{B}^{2}\;\right]}^{2}$$

Similarly, the available power transfer to the load at the maximum efficiency can be calculated from Equations (23) and (Equation 26) into (Equation 10) as:(27)$$P_{L}^{\left(b\right)}=P_{s},\max\frac{\psi^{2}+\chi^{2}}{\theta_{G}^{2}+\theta_{B}^{2}}$$

If the loses of the seawater is neglected, then the conductance, the ψ parameter, the $\theta_{G}$ parameter are zeros. Then the Equations (Equation 25) and (Equation 27) can be rewritten as:
(28a)$$\begin{array}{cc} \eta^{\left(b\right)} & =\frac{1}{2}\frac{\chi^{2}}{\;\left(\;2+\chi^{2}\;\right)} \end{array}$$
(28b)$$\begin{array}{cc} P_{L}^{\left(b\right)} & =P_{s},\max\frac{\chi^{2}}{1+\chi^{2}} \end{array}$$

Similarly, the two equations (28) are the same for the lossless medium analysis reported [17].

### 2.4. Conjugate-Image Solution

In this solution, the conjugate-image theorem [18] considers the input admittance as the conjugate of the source admittance ($Y_{\mathrm{in}}=Y_{s}^{*}$), and the output admittance as the conjugate of the load admittance ($Y_{N}=Y_{L}^{*}$). Equations (Equation 4) and (Equation 5) are identical, if the dielectric losses of the water is neglected. In contrast, if the dielectric losses is considered, then these two Equations (Equation 4) and (Equation 5) can be solved for $Y_{s}$ and $Y_{L}$. Thus, the source and load admittance can be rewritten as [14]:
(29a)$$\begin{array}{cc} Y_{s}^{\left(c\right)} & =g_{1}\;\theta_{G}+j\left(\;g_{1}\;\theta_{B}-b_{1}\;\right) \end{array}$$
(29b)$$\begin{array}{cc} Y_{L}^{\left(c\right)} & =g_{2}\;\theta_{G}+j\left(\;g_{2}\;\theta_{B}-b_{2}\;\right) \end{array}$$

If the real part of the Equation (29a) is assumed to be zero, then the expression for the maximum efficiency is obtained, which is exactly the same results of the maximum efficiency solution in the Section 2.2. Likewise, the maximum available power derived from the load is expressed in [14] is the same solution achieved in the Section 2.3.

In contrast to [14], if the real part of Equation (29a) is not neglected, which is the case (i.e., $g_{s}=g_{1}$ and $\;\theta_{G}\neq 0)$, a more precise solution can be achieved. The expression for the maximum efficiency is obtained by substituting Equation (23) into (Equation 11), giving:(30)$$\eta^{\left(c\right)}=\frac{1}{2}\frac{\psi^{2}+\chi^{2}}{{\left(\;1+\theta_{G}\;\right)}^{2}+\theta_{B}^{2}}$$

Evaluation of ${\left|\;\Delta \;\right|}^{2}$ for a given value in Equation (29) gives:(31)$${\left|\;\Delta \;\right|}^{2}={\left(\;2\;g_{1}\;g_{2}\;\right)}^{2}\;\left[\;\theta_{G}^{2}{\left(1+\theta_{G}\right)}^{2}+\theta_{G}^{2}\theta_{B}^{2}\;\right]$$

The maximum available power to the load when $g_{s}=g_{1}\;\theta_{G}\neq 0$ becomes:(32)$$P_{L}^{\left(c\right)}=P_{s},\max\frac{\psi^{2}+\chi^{2}}{\theta_{G}\;\left[\;{\left(\;1+\theta_{G}\;\right)}^{2}+\theta_{B}^{2}\;\right]}$$

Again, if the losses of the seawater are neglected, then the conductance, the ψ parameter, and the $\theta_{G}$ parameter are zeros. Then Equations (Equation 30) and (Equation 32) can be rewritten as:
(33a)$$\begin{array}{cc} \eta^{\left(c\right)} & =\frac{1}{2}\frac{\chi^{2}}{\;{\left(\;1+\sqrt{1+\chi }\;\right)}^{2}} \end{array}$$
(33b)$$\begin{array}{cc} P_{L}^{\left(c\right)} & =P_{s},\max\frac{\chi^{2}}{\sqrt{1+\chi^{2}}{\left(\;1+\sqrt{1+\chi^{2}}\;\right)}^{2}} \end{array}$$

Similarly, the two equations (33) are the same for the lossless medium analysis reported [17]. Table 1 lists a summary of the three solutions.

## 3. Calculated and Measured Results

This section provides the calculated and the measured results of the three solutions mentioned in the previous section.

### 3.1. Calculated Results

The normalized load power to the maximum available power from the source ($P_{L}/P_{s},\max$) and the efficiency of the three solutions are shown in Figure 3 as a function of ψ and χ. The results show that, for the three solutions to increase the efficiency, the χ coefficient should be increased. In particular, the mutual capacitance should be increased and the resistances should be decreased. The maximum efficiency solution gives 100% efficiency as ψ asymptotically approaches 1.

It is also clear that the power transfer capability might reduce if the system is designed to achieve maximum efficiency. This result demonstrates the reciprocal relationship between the power and the efficiency of the system. The maximum power solution can achieve 50% efficiency, but with a high power transfer range. On the other hand, the conjugate-image solution shows that if the source inductance is not zero, both the efficiency and the power transfer capability of the system decrease. However, the three solutions fail if the value of the ψ is asymptotically approaching 1.

### 3.2. Measurement Results

Figure 4 shows the experimental setup in which two pairs of square-shaped aluminum plates covered with a plastic lamination pouch for isolation were used. The coupling parameters were measured using a PicoVNA Vector Network Analyzer. The measurement was carried out over a distance (d) from 100 to 300 mm and a frequency range from 300 kHz to 1 MHz in seawater collected from the local harbor. Equation (14) is used to calculate the two coefficients ψ and χ from the measured parameters.

Figure 5 shows the two parameters ψ and χ with the change of the separation distance between the plates and the change of the frequency. Both parameters reduce with the increase in the separation distance, as the susceptance ($b_{12}$) decreases. Figure 5a shows that the ψ cannot be neglected over the frequency or the distance ranges. Thus, if the analysis for a lossless system is considered, then imprecise results are achieved.

In Figure 5b, the parameter χ decreases with the increase in the distance and the frequency ranges. As χ decreases, it is expected from Figure 3 that the efficiency will also decrease. Figure 6a depicts the change of the efficiency of the three solutions with the change of the distance and frequency. The efficiency of the system decreases with both the increase in the distance and the frequency. The maximum efficiency solution gives the highest efficiency of about 83% at 300 kHz and 100 mm. Both the maximum power and conjugate-image solutions give about half of the maximum efficiency solutions.

Figure 6 also shows the normalized power of the three solutions versus frequency and distance. The maximum power solutions have higher normalized power transfer capability as expected. The maximum normalized power at 100 mm distance and at 300 kHz is 0.994 and decreases to 0.976 at 1 MHz. The maximum efficiency solution gives a higher normalized power solution than the conjugate-image one, as shown in Figure 3. The increase in the normalized power of the later two solutions with the increase of the frequency is caused by the increase of the ψ parameter, as shown in Figure 5a.

Dionigi et al. [17] previously studied the three solutions for non-dissipative CPT systems where the dielectric of the medium is neglected. The analysis in that study works for air-gapped CPT systems but not for underwater ones. Due to the dissolved ions in the seawater, the conductivity of the water causes the losses, as shown in Figure 5b. Figure 7 depicts the relationship between the efficiency and normalized power of the underwater CPT for both dissipative (lossy) analysis and non-dissipative (lossless) analysis. It is clear that, regardless that the difference between them is small, the medium losses should be considered in the design of the system to obtain more accurate results.

## 4. Discussion

This section discusses the results of the three proposed solutions: maximum efficiency, maximum power, and conjugate-image solutions. The parameters ψ and χ are the main factors based on which the three solutions are proposed and, hence, they should be discussed thoroughly. The frequency of the electric fields and the distance between plates affect the conductance and the admittance of the underwater seawater couplers and, hence, affect the values of both ψ and χ.

### 4.1. The Parameters ψ and χ

The parameters $\psi^{2}$ and $\chi^{2}$, which are expressed in Equation (14), are equivalent to the *k*-coefficient and *Q*-factor that are defined in [16], respectively. Specifically, the parameter $\psi^{2}=k_{1}k_{2}$ where $k_{1}=g_{12}/g_{1}$ and $k_{2}=g_{12}/g_{2}$ are the normalized conductance, while the parameter $\chi^{2}=k_{1}k_{2}Q^{2}$ where *Q* is the quality factor. Although the paper used the parameters ψ and χ that are equivalent to the *k* and *Q* factors proposed in [16], the analysis in this paper provides three different solutions while the kQ-factor approach only focuses on the maximum available efficiency.

The analysis in this paper can become meaningless if negative resistances are present. For instance, if $g_{1}$, $g_{2}$ are negative the system operates as oscillator [18]. The negative terminal resistance is also possible if ψ is asymptotically approach unity, as shown in Figure 3. Specifically, $g_{12}^{2}>g_{1}g_{2}$ results in a negative $\theta_{G}$ and, hence, negative terminal resistance. Thus, the sign of the coefficient $\theta_{G}$ can be an indicator to whether or not the system is potentially an oscillator [18]. For a pure non-dissipative system with pure admittance coefficients in Equation (Equation 1), the coupling coefficient ($\kappa =Y_{12}/\sqrt{Y_{1}Y_{2}}=C_{12}/\sqrt{C_{1}C_{2}}$) is used to express the degree of coupling in the loosely coupled system. This parameter has a value between zero and unity. In such a system, the power output must equal the power input.

### 4.2. The Frequency Effect

Seawater has high conductivity caused by the dissolved salts presents as cations (Na${}^{+}$, Mg${}^{2+}$, Ca${}^{2+}$, and K${}^{+}$), and anions (Cl${}^{-}$ and SO${}_{4}^{2-}$) [19]. A water molecule is a polar molecule directed along a symmetry axis with its negative endpoints at the oxygen atom and its positive endpoints at the hydrogen atoms [20]. When the external electric fields act on the water molecules, the permanent dipoles experience torques that reorient them along the direction of the electric field. Thus, the water becomes polarized, and the polarization is proportional to the applied field [21].

The plates attract the oppositely charged ions results in the adsorption of the anions and the cations to the oppositely charged electrodes. The adsorbed ions form a layer near the surface of the electrode, known as the “Helmholtz layer”. The probability of formation of this layer in seawater is high due to the high concentration of the ions. At a low frequency, forming this layer reduces the conductance, which explains the reduction of the ψ shown in Figure 5b. However, with the increase in the frequency, the plates’ polarity changes fast, causing the dissolved ions’ fast mobility, increasing the seawater’s mutual conductivity.

Water has higher relative permittivity than air, meaning submerging the coupling capacitor in water increases the capacitance. The polarization also causes accumulations of bound charges (electron charges) within the water and on the surfaces of the electrodes. These charges produce electric fields in opposition of the applied field, which causes the increase in the mutual- and self-capacitance and, hence, the admittance. Moreover, the effects of the fringing fields increase the capacitance. Nevertheless, the rate of the increase in mutual-susceptance ($b_{12}$) is less than the product of the conductance ($g_{11}\cdot g_{22}$) with the increasing of the frequency, which results in decreasing the χ parameter, as shown in Figure 5b.

The calculated results showed that the efficiency increases with the increase of both the parameters ψ and χ, as shown in Figure 3. Thus, the efficiency can be improved by decreeing the frequency, as depicted in Figure 6a. Likewise, the maximum available power at the load increases with the parameters ψ and χ. As a result, more power can be transferred to the load at a low-frequency range if the maximum power transfer solution is followed. In contrast, if the maximum power transfer of the conjugate-image solutions is considered, then increasing the frequency improves the power transfer capability of the system, as shown in Figure 6b.

The previous studies suggested MHz frequency ranges for the CPT systems for underwater autonomous underwater vehicles [9,10,11] or underwater electric ship [12] charging applications. However, the three solutions showed that the increase in frequency could degrade the system efficiency and/or power transfer capabilities.

### 4.3. The Distance Effect

The distance between the plates at the same side (the transmitter or the receiver side) is fixed to 100 mm, as shown in Figure 4. Thus, the change conductance ($g_{11}\;\&\;g_{22}$) is almost negligible. At the same time, the distance between the transmitter and the receiver plates (d) is changed over a range of 100 to 300 mm, resulting in decreasing the conductance ($g_{12}$). As a result, the parameter ψ decreases with the increase of the transfer distance, as shown in Figure 5a. In contrast, the change in the distance has a negligible effect on the mutual susceptance ($b_{12}$), as shown in Figure 5b. As the χ is the critical parameter in calculating the available power and the system efficiency, the difference between the results of the three solutions with the change of the distance is negligible, as shown in the overlapped lines in Figure 6 and Figure 7.

## 5. Conclusions

This paper investigated a dissipative capacitive power transfer (CPT) system submerged in seawater using three solutions: maximum efficiency solution, maximum power solution, and conjugate-matching solution. The available load power and the efficiency were expressed as functions of capacitive coupling parameters. The experimental results showed that the available power decreased by the increasing of the frequency from 0.3 to 1 MHz and the separation distance change from 100 to 300 mm in the maximum available power solution. The maximum power solution gave the maximum normalized power of 0.994 at 300 kHz and a 100 mm distance. Likewise, the maximum efficiency was 83%, which was achieved using the maximum efficiency solution at the same frequency and distance. The efficiency was degraded when the frequency and the distance increased to 1 MHz. The CPT system can be a good solution for underwater wireless charging applications over a wide separation distance and low-frequency range.

## Figures and Tables

**Figure 1 sensors-21-08233-f001:**
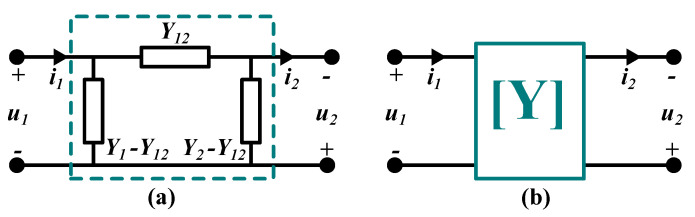
Network representation of CPT: (**a**) A π model. (**b**) A linear two-port network.

**Figure 2 sensors-21-08233-f002:**
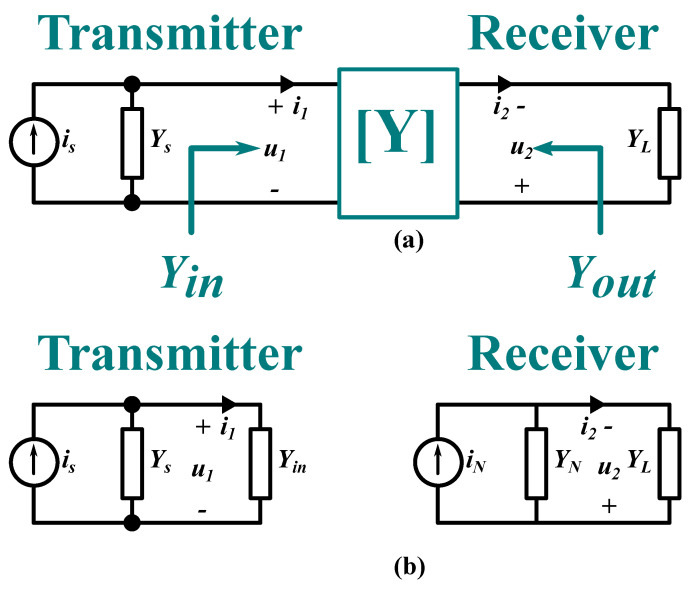
A general representation of CPT system: (**a**) two-port network connected to source and load. (**b**) The transmitter and receiver are equivalent circuits.

**Figure 3 sensors-21-08233-f003:**
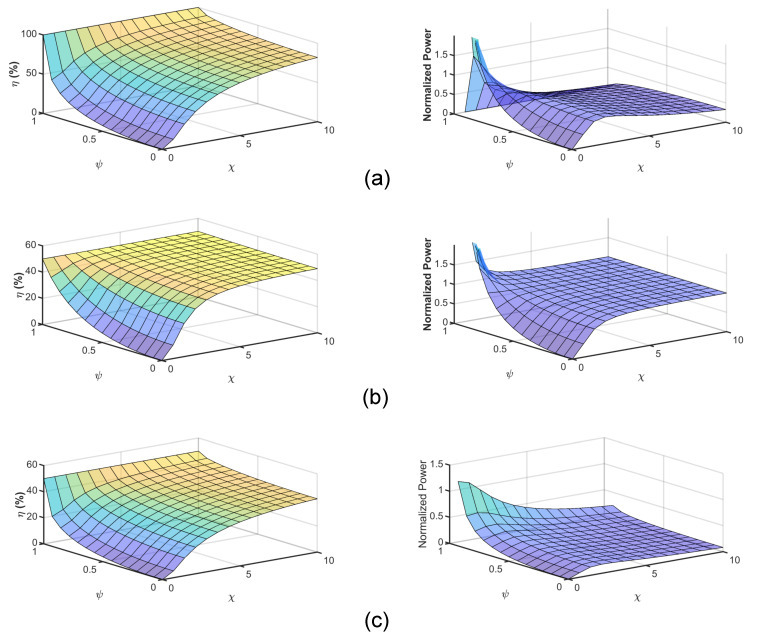
The efficiency of CPT system: (**a**) Maximum efficiency solution. (**b**) Maximum power solution. (**c**) Conjugate-image solution.

**Figure 4 sensors-21-08233-f004:**
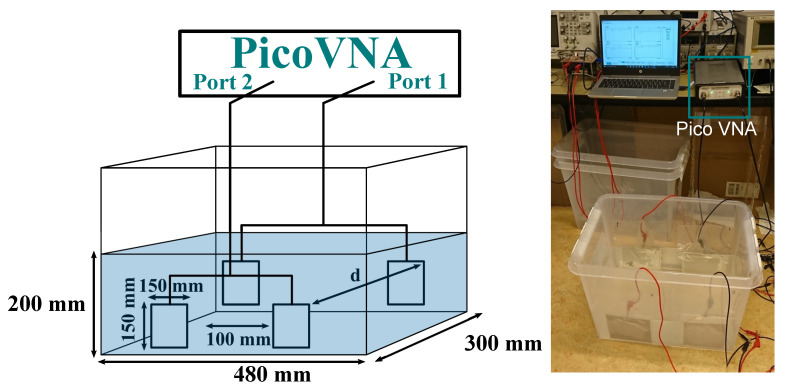
The measurement setup [14].

**Figure 5 sensors-21-08233-f005:**
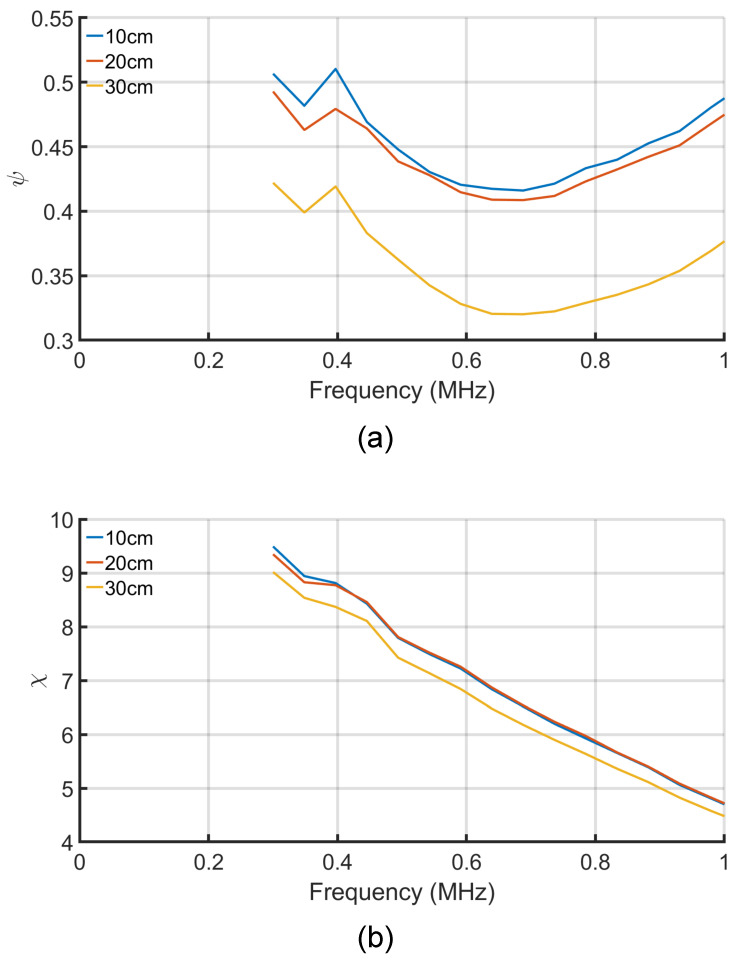
The measured ψ and χ versus the separation distance and the frequency: (**a**) The ψ coefficient. (**b**) The χ coefficient.

**Figure 6 sensors-21-08233-f006:**
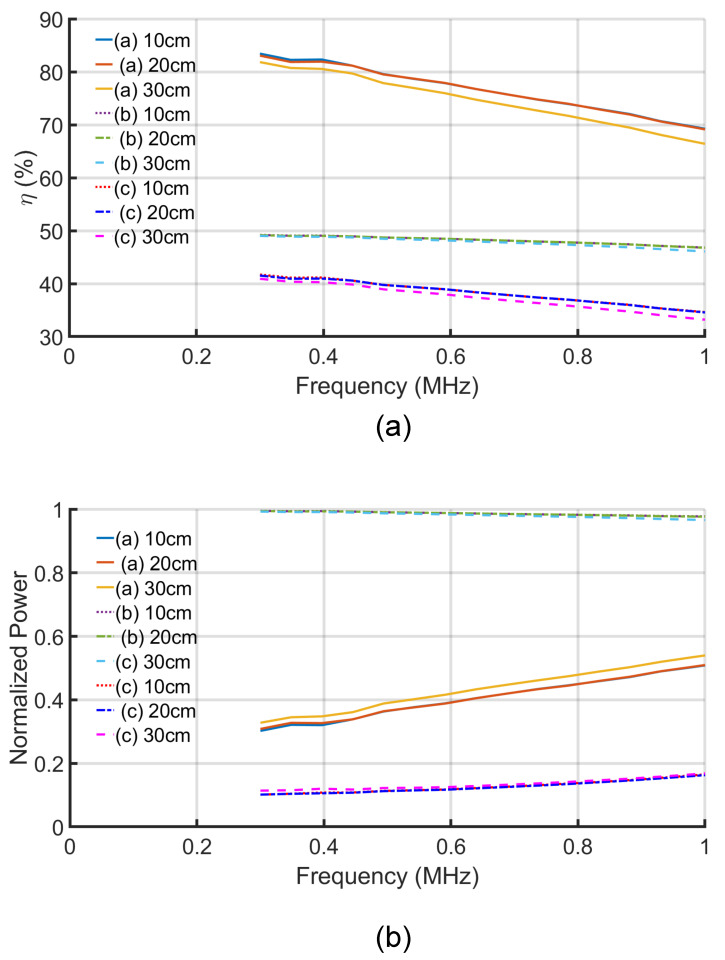
The efficiency and normalized power versus the separation distance and the frequency: (a) Maximum efficiency solution. (b) Maximum power solution. (c) Conjugate-image solution.

**Figure 7 sensors-21-08233-f007:**
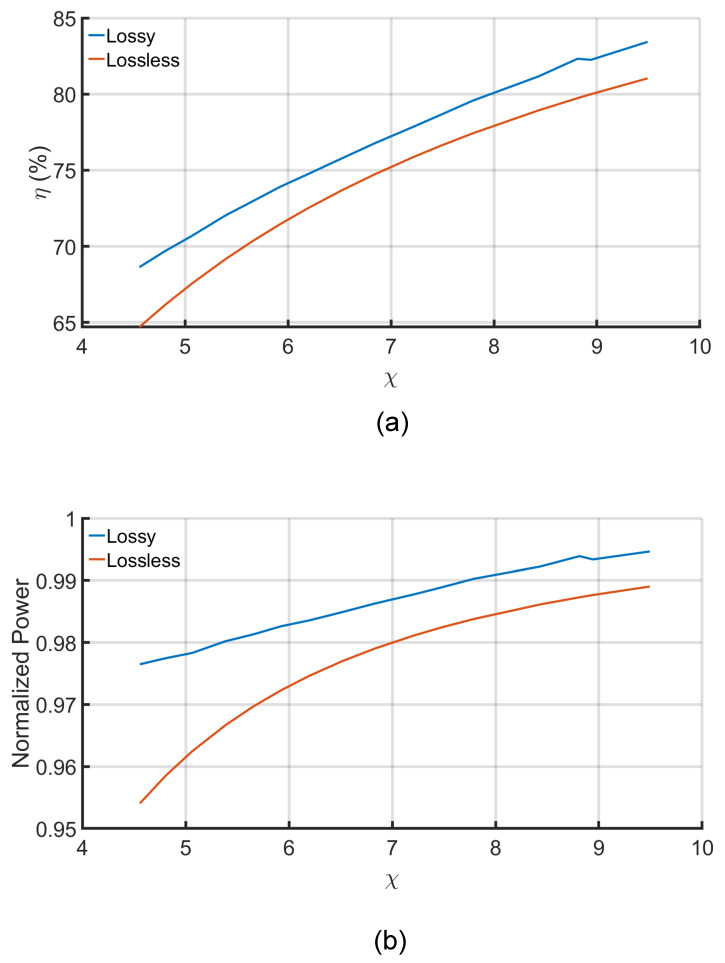
The efficiency and normalized power versus χ for lossy and lossless solutions: (**a**) Maximum efficiency solution. (**b**) Maximum power solution.

**Table 1 sensors-21-08233-t001:** A summary of the three solutions.

	Maximum Efficiency	Maximum Power	Conjugate-Image
$g_{s}$	0	0	$g_{1}\theta_{G}$
$b_{s}$	−$b_{1}$	−$b_{1}$	$g_{1}\theta_{B}$
$g_{L}$	$g_{2}\theta_{G}$	$g_{2}\left(\theta_{G}^{2}+\theta_{B}^{2}\right)$	$g_{2}\theta_{G}$
$b_{L}$	$g_{2}\theta_{B}$−$b_{2}$	$2g_{2}\theta_{B}$−$b_{2}$	$g_{2}\theta_{B}$−$b_{2}$
		**Lossy System**	
$P_{L}$	$4P_{s},\max\frac{\left(\;\psi^{2}+\chi^{2}\;\right)\;\theta_{G}}{{\left[\;\theta_{G}\left(\;1+\theta_{G}\;\right)+\theta_{B}^{2}\;\right]}^{2}+\theta_{B}^{2}}$	$P_{s},\max\frac{\psi^{2}+\chi^{2}}{\theta_{G}^{2}+\theta_{B}^{2}}$	$P_{s},\max\frac{\psi^{2}+\chi^{2}}{\theta_{G}\;\left[\;{\left(\;1+\theta_{G}\;\right)}^{2}+\theta_{B}^{2}\;\right]}$
η	$\frac{\psi^{2}+\chi^{2}}{{\left(\;1+\theta_{G}\;\right)}^{2}+\theta_{B}^{2}}$	$\frac{1}{2}\frac{\psi^{2}+\chi^{2}}{\left(\;1+\theta_{G}^{2}+\theta_{B}^{2}\;\right)}$	$\frac{1}{2}\frac{\psi^{2}+\chi^{2}}{{\left(\;1+\theta_{G}\;\right)}^{2}+\theta_{B}^{2}}$
		**Lossless System**	
$P_{L}$	$4P_{s},\max\frac{\chi^{2}}{\sqrt{1+\chi^{2}}{\left(\;1+\sqrt{1+\chi^{2}}\;\right)}^{2}}$	$P_{s},\max\frac{\chi^{2}}{1+\chi^{2}}$	$P_{s},\max\frac{\chi^{2}}{\sqrt{1+\chi^{2}}{\left(\;1+\sqrt{1+\chi^{2}}\;\right)}^{2}}$
η	$\frac{\chi^{2}}{\;{\left(\;1+\sqrt{1+\chi }\;\right)}^{2}}$	$\frac{1}{2}\frac{\chi^{2}}{\;\left(\;2+\chi^{2}\;\right)}$	$\frac{1}{2}\frac{\chi^{2}}{\;{\left(\;1+\sqrt{1+\chi }\;\right)}^{2}}$

## Data Availability

The data presented in this study are available on request from the corresponding author.

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
