# Peer review of "Optimal Solutions for Underwater Capacitive Power Transfer"

_sensors, 2021, doi:10.3390/s21248233_

Round 1

Reviewer 1 Report

I have the following comments:

1, Page 1"the challenges associated with IPT are electromagnetic interference (EMI) with communication systems and human safety issues, eddy current losses, bulky size, and high cost."I don't think the shortcomings of IPT system analyzed by the author are completely correct. Please provide more detailed reference.

2, I think the author's description of the innovation of this paper is not clear enough. Please describe it in detail.

Author Response

We want to thank the reviewer for the valuable feedback on the manuscript. We incorporated changes to reflect most of the suggestions provided by the reviewers and highlighted the changes within the manuscript in red font. Here is our response to the reviewers' comments:

Comment 1: Page 1 "the challenges associated with IPT are electromagnetic interference (EMI) with communication systems and human safety issues, eddy current losses, bulky size, and high cost."I don't think the shortcomings of IPT system analyzed by the author are completely correct. Please provide more detailed reference.

Response: Thank you for pointing this comment, we agree about this point. Therefore, we provide two references that discuss the shortcomings of IPT in more detail.

Comment 2: I think the author's description of the innovation of this paper is not clear enough. Please describe it in detail.

Response: We also agree about this and accordingly modified the introduction to emphasize the main contribution of the paper and the difference between the analysis in the paper and the literature.

Reviewer 2 Report

The paper is very very similar (the experiment is the same!) to the paper published in the IEEE as Proceedings of IEEE WPTC 2021:

  1. Mahdi, B. Hoff and T. Østrem, "Maximum Available Power of Undersea Capacitive Coupling in a Wireless Power Transfer System," 2021 IEEE Wireless Power Transfer Conference (WPTC), 2021, pp. 1-4, doi: 10.1109/WPTC51349.2021.9458006.

So the originality of the manuscript is very low.  In the opinion of this reviewer the manuscript should not pass tests to check for plagiarism and multiple publication. If Sensor Journal does not care about the plagiarism with an already published paper of the same authors, the manuscript is well written and could be of interest for readers, even if the mathematical treatment of traditional wireless capacitive coupling is quite standard and already well established in the technical literature, as can be observed in the cited papers.

Another remark of this reviewer is on the use of  parameters fi and ki (Greek symbols in the manuscript) used by the authors to present the main results. These parameters  are not easily referred to electrical quantities, therefore the understanding of the results is not immediate and the manuscript is not very significant.

Other remarks are listed in the following

  • The load admittance Y_L is not defined.
  • At row 78 the voltage u2 is the voltage drop across the load Y_L produced by the current i2, so the direction of u2 is not clear.
  • Equation (4) seems to be not correct since the second row of current vector should be zero and not i2. Therefore it is not necessary to apply an open circuit condition but only to solve the equations through nodal analysis.
  • The power Ps in (10) is not defined.
  • The statements in lines 97-103 should be changed by omitting “determine the value of the source” as the considered system is linear.
  • The titles of references [12] and [13] are identical!
  • There are several typos in the text that need to be corrected.

Author Response

We want to thank the reviewer for the valuable and thorough feedback on the manuscript. We incorporated changes to reflect most of the suggestions and comments provided by the reviewers. We highlighted the changes in the manuscript in red font.

Our response to the reviewers' comments and concerns is as follows:

Comment 1: 

The paper is very very similar (the experiment is the same!) to the paper published in the IEEE as Proceedings of IEEE WPTC 2021:

  1. Mahdi, B. Hoff and T. Østrem, "Maximum Available Power of Undersea Capacitive Coupling in a Wireless Power Transfer System," 2021 IEEE Wireless Power Transfer Conference (WPTC), 2021, pp. 1-4, doi: 10.1109/WPTC51349.2021.9458006.

So the originality of the manuscript is very low.  In the opinion of this reviewer the manuscript should not pass tests to check for plagiarism and multiple publication. If Sensor Journal does not care about the plagiarism with an already published paper of the same authors, the manuscript is well written and could be of interest for readers, even if the mathematical treatment of traditional wireless capacitive coupling is quite standard and already well established in the technical literature, as can be observed in the cited papers.

Response:
Regarding the multiple publications, this paper is an extension of our contribution to IEEE 2021 Wireless Power Transfer Conference (Mahdi, B. Hoff and T. Østrem, "Maximum Available Power of Undersea Capacitive Coupling in a Wireless Power Transfer System," 2021 IEEE Wireless Power Transfer Conference (WPTC), 2021, pp. 1-4, doi: 10.1109/WPTC51349.2021.9458006.) The paper is selected to submit to the special issue "Selected Papers from 2021 IEEE MTT-S Wireless Power Transfer Conference" in MDPI Sensors. Thus, we expanded the analysis and provided more discussions.

The conference manuscript proposes the maximum available power using the conjugate-image solution only. In contrast, this manuscript presents another two solutions, namely, the maximum efficiency and the maximum power solutions. Besides, the conjugate-image solution in this paper is adjusted to achieve more accurate results, as highlighted in lines 160 to 166 of the attached manuscript.

Regarding the plagiarism, we presented all the previous works in the literature that we used in the analysis to avoid publishing someone else's as our ones. Besides, we cited our own works, specifically the conference paper, to avoid self-plagiarism. The authors care about academic integrity and are eager not to cheat or violate it. Thus, the conference manuscript was cited in different parts of this paper.

Comment 2: the use of parameters fi and ki (Greek symbols in the manuscript) used by the authors to present the main results. These parameters are not easily referred to electrical quantities, therefore the understanding of the results is not immediate and the manuscript is not very significant.

Response: Thank you for pointing this out. We agree about this point and accordingly, we modified the discussion by adding section 4.1 (lines 234 - 251) that presents the physical meaning of these two coefficients and connected them to the previous literature.

Comment 3: The load admittance Y_L is not defined.

Response: We agree and revised line 84 of the attached manuscript accordingly.

Comment 4: At row 78 the voltage u2 is the voltage drop across the load Y_L produced by the current i2, so the direction of u2 is not clear.

Response: Agree. We modified lines 80-83 based on the argument in lines 77-78 of the attached manuscript.

Comment 5: Equation (4) seems to be not correct since the second row of current vector should be zero and not i2. Therefore it is not necessary to apply an open circuit condition but only to solve the equations through nodal analysis.

Response: Agree. The short circuit condition is followed only to achieve equation 5. The equations can also be derived through nodal analysis which is proposed in the literature.

Comment 6: The power Ps in (10) is not defined.

Response: Agree. We added equation 8 to the attached manuscript in lines 98-99.

Comment 7: The statements in lines 97-103 should be changed by omitting “determine the value of the source” as the considered system is linear.

Response: We agreed with this and we revised lines 107-108 of the attached manuscript, accordingly.

Comment 8: The titles of references [12] and [13] are identical!

Response: We modified the manuscript accordingly.

Comment 9: There are several typos in the text that need to be corrected.

Response: We agree with this point and revise the manuscript to avoid them.

Reviewer 3 Report

Implementing underwater capacitive power transfer is a fantastic idea. Although capacitive power transmission has a limited number of applications, underwater power transfer appears to be an intriguing subject.

The experiment was carried out using static seawater, which differs from the results of the lossless analysis. However, I'm curious whether the authors might remark on what occurs when dynamic water instead of static water is employed.

For additional exploration, multiphysics simulations can be run. Why is figure 4 positioned above figure 3? This should be rectified.

Author Response

We appreciate the time and effort that the reviewer dedicated to providing us with valuable feedback and encouraging comments on the manuscript. We have incorporated changes to reflect the suggestion and recommendations provided by the reviewers. The changes in the manuscript are highlighted in red font.

Here we respond to the reviewers' comments:

Comment 1: The experiment was carried out using static seawater, which differs from the results of the lossless analysis. However, I'm curious whether the authors might remark on what occurs when dynamic water instead of static water is employed.

Response: We would like to thank you for this brilliant suggestion. It would have been interesting to explore this issue. However, the test is carried out for static water due to the limitation of the experiment setup. We tried to stir the water in the container and check the results, but it had negligible effects. Nevertheless, we need to investigate the results over different flow rates, which require a new experimental setup. And because of the limited time of answering the reviewers, we could not provide the effect of the dynamic water. We will investigate it soon.

Comment 2: For additional exploration, multiphysics simulations can be run.

Response: Thank you again for this suggestion. It is an interesting idea to compare the experimental and calculated results with numerical results using multiphysics simulation. However, as this manuscript is an expansion of the conference paper, we tried to expand the conference manuscript and make the results in this paper comparable to those in the conference paper. We have another article that we will submit soon. In this article, we analyze the underwater CPT using analytical, FEM, and measured values.

Comment 3: Why is figure 4 positioned above figure 3? This should be rectified.

Response: Thank you for pointing this out. We change the manuscript accordingly.

Round 2

Reviewer 1 Report

Thanks for the revision, I have no more questions.

Reviewer 2 Report

This reviewer is satisfied with the modifications made by the author in the revised version of the manuscript